# Improving Automatic Smartwatch Electrocardiogram Diagnosis of Atrial Fibrillation by Identifying Regularity within Irregularity

**DOI:** 10.3390/s23229283

**Published:** 2023-11-20

**Authors:** Anouk Velraeds, Marc Strik, Joske van der Zande, Leslie Fontagne, Michel Haissaguerre, Sylvain Ploux, Ying Wang, Pierre Bordachar

**Affiliations:** 1Cardio-Thoracic Unit, Bordeaux University Hospital (CHU), F-33600 Bordeaux, France; anoukvelraeds@gmail.com (A.V.); joske.van.der.zande@gmail.com (J.v.d.Z.);; 2IHU Liryc, Electrophysiology and Heart Modeling Institute, Fondation Bordeaux Université, F-33600 Bordeaux, France; 3Biomedical Signals and Systems, TechMed Centre, University of Twente, 7522 NH Enschede, The Netherlands

**Keywords:** atrial fibrillation, Apple Watch, algorithm, electrocardiography, mobile health, wearables, irregularity, regularity, smartwatch

## Abstract

Smartwatches equipped with automatic atrial fibrillation (AF) detection through electrocardiogram (ECG) recording are increasingly prevalent. We have recently reported the limitations of the Apple Watch (AW) in correctly diagnosing AF. In this study, we aim to apply a data science approach to a large dataset of smartwatch ECGs in order to deliver an improved algorithm. We included 723 patients (579 patients for algorithm development and 144 patients for validation) who underwent ECG recording with an AW and a 12-lead ECG (21% had AF and 24% had no ECG abnormalities). Similar to the existing algorithm, we first screened for AF by detecting irregularities in ventricular intervals. However, as opposed to the existing algorithm, we included all ECGs (not applying quality or heart rate exclusion criteria) but we excluded ECGs in which we identified regular patterns within the irregular rhythms by screening for interval clusters. This “irregularly irregular” approach resulted in a significant improvement in accuracy compared to the existing AW algorithm (sensitivity of 90% versus 83%, specificity of 92% versus 79%, *p* < 0.01). Identifying regularity within irregular rhythms is an accurate yet inclusive method to detect AF using a smartwatch ECG.

## 1. Introduction

One of the most common cardiac arrhythmias is atrial fibrillation (AF), which results in a completely irregular ventricular rhythm on the electrocardiogram (ECG), due to complete loss of organized atrial contractility with signals being redirected intermittently to the ventricles [1,2]. By definition, AF is an irregular rhythm without any pattern of regularity, while most other irregular rhythms express regular patterns among the irregularities [3]. AF may be asymptomatic, but many patients experience palpitations, fatigue, lightheadedness, or even syncope [2]. The absence of symptoms leads to a late diagnosis of the arrhythmia with a higher risk of clinical complications [4]. AF increases the risk of stroke and heart failure and, therefore, it is imperative to diagnose the arrhythmia in a timely manner [5]. This is why wearables with AF screening capabilities are gaining in popularity [6]. Apple Inc. was the first to obtain FDA approval for the automatic detection of AF on the ECG of a smartwatch [7]. The growing number of consumers with smartwatches that can self-record and auto-diagnose AF is associated with a growing number of diagnoses in the early stages of the disease. Early treatment is more effective in AF, reducing the risk of stroke and heart failure [3]. 

Nevertheless, the automatic diagnosis of AF in today’s smartwatches is far from perfect. Only a specific range of heart rates is allowed for the classification of AF, varying from 50 to 120 or 150 beats per minute [8]. In addition, the Apple Watch (AW) diagnosis is often inconclusive, which may delay the diagnosis significantly [9,10]. It is also possible to obtain a poor reading notification, which arises due to electrode contact issues or movement, etc. In an early study, we reported that when inconclusive results are considered false results, there are important decreases in sensitivity (87% vs. 99%) and specificity (86% vs. 93%) [9]. While in this study, only AF and sinus rhythm were compared, other ECG abnormalities should also be taken into account for the validation of the AW classification. More recently, we confirmed the importance of coexisting ECG abnormalities with associated decreases in sensitivity (69%) and specificity (81%) [10]. It Is not surprising that coexisting ECG abnormalities have an important negative impact on accuracy as other arrhythmias (e.g., premature contractions) may also result in an irregular rhythm, leading to false positives. Also, AF is increasingly prevalent with age and the Apple Heart Study contained mostly young and healthy patients [6]. The issue of overdiagnosis can arise when smartwatch ECGs exhibit low sensitivity and specificity in detecting AF. This scenario may result in false negatives and false positives, leading to the misclassification of individuals [11].

Therefore, automatic detection of AF through lead I smartwatch ECGs seems to be promising but could still use some improvements [12]. Our study aims to develop an improved automatic AF detection algorithm for patients with or without coexisting ECG abnormalities. We propose an inclusive two-step approach; first identifying any irregular rhythm and then excluding ECGs which show patterns of regularity [13]. This approach aims to be more inclusive (no rejection of low/high heart rates or less quality tracings) and aims to reduce the number of false positives. 

## 2. Material and Methods

### 2.1. Patient Population

The dataset contained ECGs of 723 patients hospitalized in the Cardiology department; 21% had AF and 24% had an ECG without any abnormality [10]. At rest conditions, participants underwent a standard 12-lead electrocardiogram, followed by a smartwatch ECG recording using the Apple Watch Series 5 (Apple Inc., Cupertino, CA, USA). The first completed smartwatch ECG tracing was used. The 12-lead ECG diagnosis, as it was interpreted by an expert electrophysiologist, was used as the gold standard for the diagnosis of the concomitant smartwatch ECG, also confirmed by an expert electrophysiologist. Our institutional review board authorized the research, and the participants gave written informed consent.

### 2.2. Data Arrangement

Digital ECG signals were extracted from the PDF files received from the AW after recording. The PDF files were transformed into SVG files in Python^®^ (version 3.9.18) and digitalization was performed using a vector detection approach, based on the red RGB code of the signal on the PDF of the AW ECG [14]. The ECGs were then imported into MATLAB^®^ (version 9.12.0.1927505) together with the expert 12-lead ECG diagnoses and automatic AW diagnosis.

### 2.3. R Peak Detection

The first step was the detection of the R waves. The detection of the R peaks was performed in a multi-step approach, where the R peaks were enhanced by the Maximal Overlap Discrete Wavelet Transform (MODWT) from the Wavelet Toolbox from MATLAB^®^ (version 9.12.0.1927505) and the find peaks function from the Signal Processing Toolbox from MATLAB^®^ (version 9.12.0.1927505) was used in three steps to detect the R peaks [15].

### 2.4. Irregularity

The exact mechanisms of the AW algorithm for the detection of AF are unknown, but it is presumably based on the irregularity of ventricular complexes [8]. In our approach, we also used this feature as the first step of the novel algorithm [13].

Two methods were used to check for irregularity. The first method summed every *RR* interval that is equal to the median of the *RR* intervals of the ECG with a range of 15 milliseconds, this is then divided by the heart rate (Equation (1)). The second method used is a Singular Value Decomposition (SVD) that is performed on the Lorenz plot of the *RR* intervals. The singular values are the values that give the lengths of the two main directions of the Lorenz plot. These singular values were divided by each other, and this gave a ratio in singular values (Equation (2)). A combination of both features was used for AF detection and thresholds were found by creating a Receiver Operating Characteristic (ROC) curve, which was generated with the different combinations of threshold values for both features. A calculation of the distance from the perfect point (0, 1) on the ROC curve was performed to obtain the best threshold values [16].
(1)$$for\;RR=median\left(RR\right)\pm 15\;\mathrm{ms}\;:\;\;count\;RR=\frac{sum\left(RR\right)}{heart\;rate}$$
(2)$$SVD\;ratio=\frac{singular\;value\;1}{singular\;value\;2}$$

### 2.5. Finding Regularity in Irregularity

When observing Lorenz plots of irregular ECGs that are not AF, patterns of regularity may be identified. In the case of premature complexes, three types of *RR* intervals can be appreciated: the interval between normal beats, the short interval following the normal beat until the premature complex, and the long interval between the premature complex and the following normal beat. By comparing the Lorenz plot of a patient with AF with a patient with premature atrial complexes (PACs) (Figure 1), clusters of points can easily be identified in the latter patient, which signify the three types of *RR* intervals. Both ECGs showed an irregular rhythm according to the irregularity feature.

We used the k-means clustering function from the Statistics and Machine Learning Toolbox from MATLAB^®^ (version 9.12.0.1927505) to quantify the observed clustering [17]. This clustering method was chosen due to its computational efficiency, ease of implementation, and suitability for the dataset’s characteristics. It is also clear and straightforward, ideal for this research where simplicity and clearness are valued over unnecessary complexity. In addition, this function is robust to noise, as clusters are found in a certain radius from a centre point and it is very useful in well-divided clusters, as is the case in this research. K-means clustering is able to implement distance and width conditions for the clusters. These two conditions were included to prevent the accidental detection of clusters in an irregularly irregular signal (Equation (3)). The number of clusters found signifies the presence or absence of other irregular ECG anomalies.
(3)$$\begin{array}{c} for\;distance>0.4\\ for\;cluster\;width\;\leq \;0.5 \end{array}\}\begin{array}{c} \;\;\left[idx,\;C\right]=kmeans\left(2D\;list\;RR,\;k\right)\\ \;\;cluster\;count=sum\left(idx\right) \end{array}$$

### 2.6. Validation

A systematic approach was used to find the optimal threshold values by exploring various combinations of the features and selecting the combination that minimizes the Euclidean distance from the ideal point (0, 1) on the ROC curve [16]. These threshold values were then used for the test set. Sensitivity, specificity, PPV, NPV, F2 score, and accuracy will show the validity of the algorithm based on the test set. The F2 score is based on the idea that sensitivity should be given more weight than PPV [18].

The diagnostic accuracy of the new algorithm was then compared with the diagnostic accuracy of the existing AF detection algorithm of the AW [10]. Inconclusive results were considered as false results [9,10].

The McNemar test was used to identify if there is a significant difference between the proposed algorithm and the existing AW algorithm [19]. The null hypothesis was that the created algorithm and the AW algorithm had the same detection validation of AF. The alternative hypothesis was that a significant difference was found between the two tested algorithms. This was tested through the χ^2^-test on the discordant values of the two tests with 1 degree of freedom. If a *p*-value lower than 0.01 was found by performing this test, the null hypothesis was rejected and a significant difference between the two algorithms was proven.

## 3. Results

### 3.1. Baseline Characteristics

Table 1 summarizes the baseline patient characteristics. The study’s 723 participants underwent simultaneous standard 12-lead and AW recordings [10]. In total, 173 (24%) of the subjects were without known cardiac disease and 154 (21%) had been diagnosed with AF. There were no mismatches (presence or absence of AF) between the expert diagnosis of the 12-lead ECG and the Apple Watch ECG in any of the patients. The automatic diagnosis of the AW declared 137 ECGs (19%) as AF and 142 ECGs (19%) as inconclusive. Examples of the Apple Watch (AW) ECG, one without ECG abnormalities and one from an AF patient, can be seen in Figure 2, respectively.

### 3.2. Irregularity

For the first tested irregularity feature, where *RR* intervals within a range of 15 ms were divided by the heart rate (count RR), the median of the ECGs without anomalies was 0.269. For AF ECGs, the count *RR* median was 0.059. For atrial extrasystoles, this was 0.188, and for ventricular extrasystoles, 0.154. These and values of other irregular arrhythmias can be found in Table 2 (a). 

The second method attempted was the Singular Value Decomposition, which evaluated the ratio between the two singular values of the Lorenz plot of *RR* intervals (SVD ratio). A regular ECG should give us a higher ratio than an irregular ECG. The median of the ECGs without anomalies was 64.41. For AF ECGs, this was 7.45. For atrial extrasystoles, this is 7.36, and for ventricular extrasystoles, 15.80. These and values of other irregular arrhythmias can be found in Table 2 (b). 

The optimal threshold values for the combined use of the SVD ratio was ≤ 13.31 and for the count *RR* ≤ 0.146 for AF detection, these were found by creating a ROC curve of the combinations of both thresholds and finding the closest point to the perfect point (0, 1). By calculating the validation values of the test set using these threshold values that both need to hold true, a sensitivity of 89.66% is found, a specificity of 86.96%, a PPV of 63.41%, an NPV of 97.09%, an F2-score of 82.80%, and the accuracy is 87.50%.

### 3.3. Finding Regularity in Irregularity

After classifying irregularity, irregular ECGs were analyzed for the presence of clustering in the Lorenz plot. As can be seen from Table 2, most AV blocks were already diagnosed as no AF by the irregularity criterion, so the second function was mainly used for premature beat detection. A few examples of the cluster search on the Lorenz plots are shown in Figure 3 of, respectively, a normal ECG (a), AF (b), an ECG with PACs (c), and with premature ventricular complexes (PVCs) (d). The optimal threshold was found by adjusting the parameter in irregular ECGs with and without AF.

### 3.4. Validation

The hypothesis of adding the finding clusters feature to the algorithm was to improve the specificity of the irregularity feature solely. Validation of the proposed algorithm was performed by combining the three thresholds that must hold true to diagnose AF for each of the 144 ECGs in the test set; the SVD ratio needed to be ≤13.31, the count *RR* needed to be ≤0.158, and the cluster count needed to be equal to 1.

Using these thresholds, the following diagnostic values were found by the test set: a sensitivity of 89.66%, a specificity of 92.17%, a PPV of 74.29%, an NPV of 97.25%, an F2 score of 86.09%, and an accuracy of 91.67%.

The AW algorithm was also validated on the test set, with the inconclusive results taken as false results. This gives a sensitivity of 82.76%, a specificity of 79.13%, a PPV of 50.00%, an NPV of 94.79%, an F2 score of 73.17%, and an accuracy of 79.86%. For the 25 inconclusive AW results in the test set, 19 were correctly identified by the novel algorithm.

The McNemar test was used to check for a significant difference between the two algorithms. The discordance is found in the cases where one of the algorithms correctly identifies AF and the other does not. The χ^2^-test was performed on the discordance and the *p*-value was calculated on this test. A *p*-value of 0.0014 was found, which rejects the null hypothesis and proves a significant difference between the two algorithms.

Using the novel algorithm, nine false positives occurred, one had an ECG without abnormalities, while the others had abnormalities such as premature beats, abnormal QRS or flutter/atrial tachycardia. In comparison, the AW algorithm resulted in 24 false positives.

## 4. Discussion

We present a novel algorithm which automatically detects AF in a large and challenging group of patients. Where the AW does not look further than irregularity for AF detection, we showed that improved AF detection needs a second step. Identifying clusters of regularity within the rhythms declared as irregular is highly effective. Using cluster identification in Lorenz plots increased the diagnostic accuracy as compared with the pre-existing automatic diagnostic algorithm within the smartwatch. It is challenging to compare existing studies of AF detection using the Apple Watch ECG as the type of subject and clinical context in which the tracings are acquired may be different or even unknown. The first, and perhaps most important study performed, by Apple remains unpublished, but the main results are found within the FDA clearance of 2019 (21 CFR 870.2345) and can be found in Table 3. It is unclear how the subjects were recruited and how the smartwatch ECGs were registered. A later study performed by our group which aimed to compare three different smartwatches found similar results for the Apple Watch as described in the FDA report (Table 3), but half of our patients had normal ECGs and the other half had AF, which is also not a very realistic setting [9]. In our most recent and much larger study, we also included patients with co-existing ECG abnormalities [10]. The accuracy was notably lower with also double the number of patients where no diagnosis given by the smartwatch (19% vs. 9% in the FDA report). The current work proposes an algorithm which outperforms previous results by detection of clusters but also by an all-inclusive approach.

Automatic AF detection through smartwatches leads to four times earlier detection of AF, allowing earlier and more efficient treatment [6]. The higher sensitivity of the novel algorithm improves the accurate diagnosis of AF and provides more patients with an early diagnosis and, thereby, more effective treatment. Moreover, through the enhancement of specificity through the novel algorithm, the occurrence of false positives is mitigated, resulting in a reduced number of erroneously diagnosed patients entering the diagnostic process.

Current smartwatches are associated with a high rate of non-diagnoses due to the algorithm returning non-conclusive or out-of-range results. These labels greatly decrease the confidence in using wearables as a diagnostic tool. Efforts need to be made to reduce the amount of rejected ECGs to a minimum and the use of more inclusive algorithms is an important step to the usability of smartwatch ECGs. The proposed algorithm is all-inclusive, decreasing the uncertainty smartwatch users may be confronted with when being presented with inconclusive ECGs. While AW classified a significant number of ECGs inconclusive, this was not the case in the novel algorithm as all ECGs were included in the algorithm. Including all the ECGs assured the higher validity and usability of the novel algorithm. The novel algorithm correctly identified most of the ECGs which were classified as inconclusive by the AW algorithm (19/25). The obvious potential downside of a more inclusive algorithm is the higher risk of false positives, but this was not the case with the novel algorithm because despite not rejecting any ECG, the specificity was still higher. 

Combining a higher inclusivity with a higher validity provides a more accurate diagnosis of all patients using a smartwatch for the detection of AF. These patients may receive early treatment, which has been shown the most effective in AF patients and the reduction of complications due to an undiagnosed disease [20].

Apparently, the feature that counts *RR* intervals of the same length is already able to exclude some regular irregular cases. This can be seen in Table 2 (a) where, for example, PACs have a higher median for this feature and do not surpass the threshold value for AF detection.

Detecting clusters within Lorenz plots has proven to be feasible and adding this feature improved the specificity of the algorithm by 4 percent points. It should be noted that only the Lorenz plots of the irregular cases were used because our approach was to exclude the false positives from the ECGs suspected of showing AF. Using this feature, only a single patient was misdiagnosed as AF; while in fact, the patient had PVCs which made the rhythm irregular. No patients were misdiagnosed for ECGs with PACs or AV blocks. The question arose whether an AF ECG with PVCs or PACs would also show clustering, but this was not the case in any of the ECGs with either of these two pathologies.

Figure 4 shows a challenging smartwatch ECG in a patient in sinus rhythm who had both PACs and PVCs in a single tracing. Without surprise, the AW algorithm gives the wrong diagnosis of AF. The novel algorithm reports a count *RR* of 0.053 and an SVD ratio of 5.58, which both declare the rhythm as irregular, suspect of AF. However, when using the second feature which searches for clusters in the Lorenz plot, clusters were found (Figure 4b). As the algorithm shows, if a regularity in the irregularity is found, the diagnosis of AF could be successfully rejected.

### 4.1. Future Approaches

The ECG application of selected Withings smartwatches (Scanwatch) recently added a feature which displays QRS, PR, QT, and QTc intervals on the ECG report [21]. This signifies that these smartwatches identify P waves, which also is a sign of the absence of AF [3]. However, this new feature remains untested, so the validity is unknown. We have attempted the detection of P waves in our dataset but found it challenging and did not report it in this work. If it shows to be a valid method for P wave detection, it could also be used in AF detection to exclude these cases from AF. 

Future approaches may include machine learning methods, using features beyond clinical comprehension. Machine learning is well adapted for ECG diagnostics and may outperform the algorithms proposed in this paper well [22,23]. A systematic review on this topic shows very high performance with a mean sensitivity of 94.80% and a mean specificity of 96.96% in 26 studies of ML in AF detection on smartwatch ECGs [24]. ML is, thus, very robust for easy pattern recognition such as the presence or absence of AF on a 30 s smartwatch ECG. However, ML requires immense computing power and is not available in offline wearables. ML approaches can only be used to adjudicate a diagnosis by use of an online platform. As smartwatches need to be able to directly deliver a diagnosis (absence or presence of AF), a data science approach is still required. While ML hold enormous potential for correction or confirmation of traditional algorithms, the limitations of current on-board algorithms inhibit the widespread use and acceptance of AF screening with a smartwatch ECG, let alone more complicated diagnoses such as other arrhythmia or conduction diseases.

### 4.2. Study Limitations

We only recorded one AW ECG per patient. In reality, when one may obtain an inconclusive, too high/low heart rate, or poor reading notification, the person can repeat the ECG and receive a more complete diagnosis. However, repeating tests using an imperfect algorithm will result in multiple diagnoses, damaging the confidence of the user and decreasing the clinical applicability of the smartwatch for the use of AF detection.

While our data holds relatively many subjects with cardiac abnormalities, a sub-validation for specific pathologies (PACs, PVCs, AV block) remains challenging to merit a proper subgroup validation.

We analysed a dataset with patients from the Cardiology department, which is not representative of reality. The prevalence of ECG anomalies is, therefore, much higher than in a randomly taken patient group. Therefore, it would be interesting to also test this algorithm on a dataset that is a better representation of reality. 

R wave detection showed some difficulties in some cases where the QRS complex did not resemble a normal QRS complex, as the method used looks for a wavelet that matches a reference QRS complex. This was surely the case for pacemaker and CRT patients. However, AF detection in these cases has very limited clinical relevance, as patients with CRT or a pacemaker are continuously being monitored for arrhythmias through their implanted device [25]. Two of the false positives found are in pacemaker and CRT patients, so, excluding these cases would improve the performance of the novel algorithm and of the AW.

The Lorenz plot interpretation was performed very carefully, avoiding the detection of clusters within AF. Therefore, the threshold values for minimum distance between clusters were taken very widely to prevent accidentally detecting clusters in AF. Thus, in some cases, clusters were missed. An example of an ECG with undetected PVCs can be seen in Figure 5. However, in our study, there was only one ECG in the test set with PACs, PVCs, or AV block which was misdiagnosed as AF.

## 5. Conclusions

We developed and validated a multi-step algorithm that correctly identifies AF in a complex group of patients, outperforming the existing algorithm. We first identified irregularity using Lorenz plots and, in a second step, identified regularity within irregularity using clusters. Future research will show to what extent this can be used in AF detection.

## Figures and Tables

**Figure 1 sensors-23-09283-f001:**
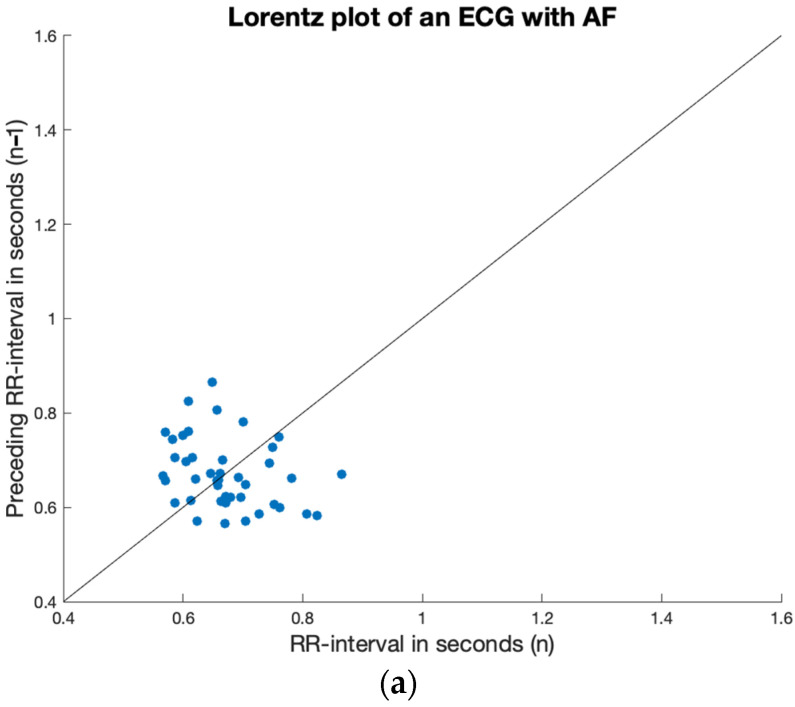
The Lorenz plot of a patient with AF (**a**) and a patient with PACs (**b**). The diagonal line shows the line along which the perfectly regular intervals would be.

**Figure 2 sensors-23-09283-f002:**
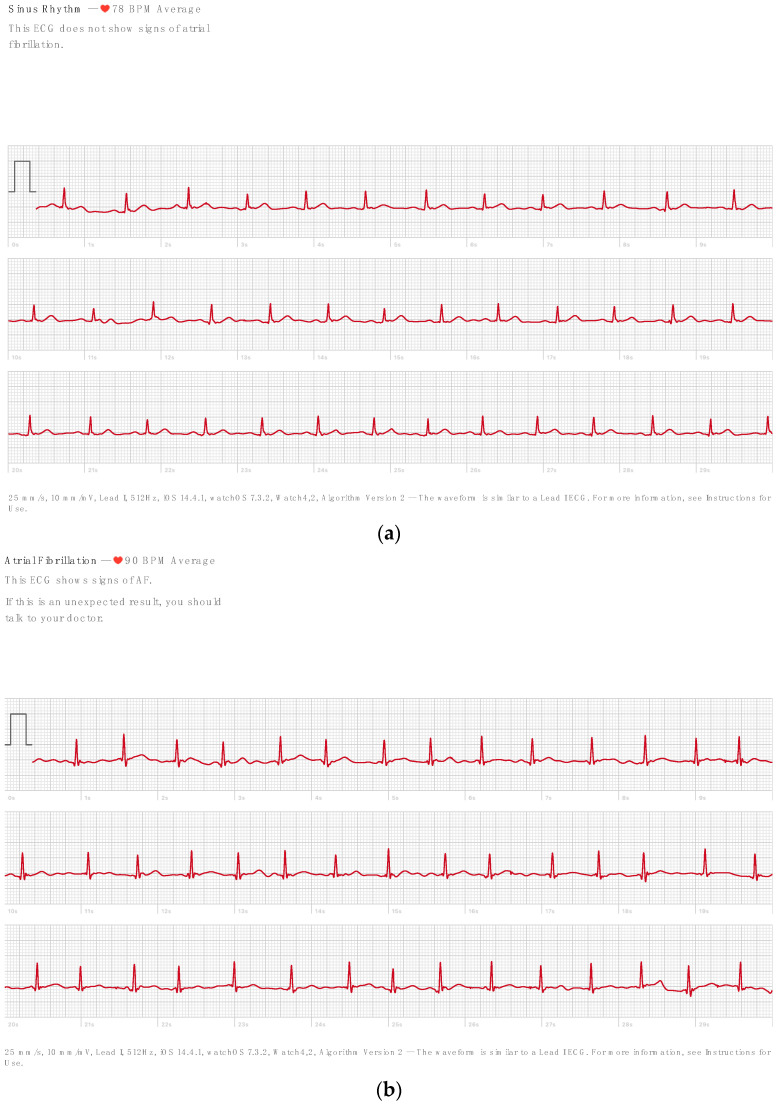
Example of the Apple Watch ECG without abnormalities (**a**) and from an AF patient (**b**) [10].

**Figure 3 sensors-23-09283-f003:**
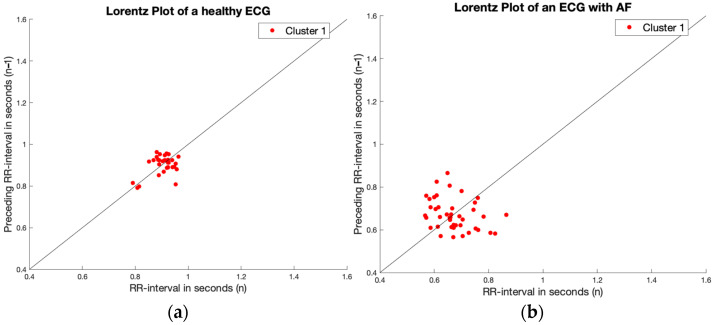
The K-means clustering function in four different ECGs; (**a**) is a healthy ECG without any irregularity and clusters, there are also no clusters found; (**b**) is an ECG with AF with irregularity, but no clusters, which are also not found by the function; (**c**) is an ECG with PACs, so with irregularity and clusters are found; and (**d**) is an ECG with PVCs, so also with irregularity and clusters are found.

**Figure 4 sensors-23-09283-f004:**
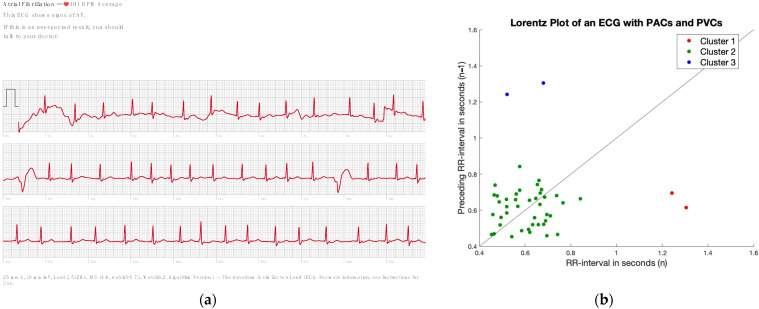
The patient is diagnosed with AF by the AW algorithm (**a**). However, the novel algorithm shows by finding clusters in the Lorenz plot (**b**), regularity within the irregularity, and so the novel algorithm correctly identifies this patient with no AF.

**Figure 5 sensors-23-09283-f005:**
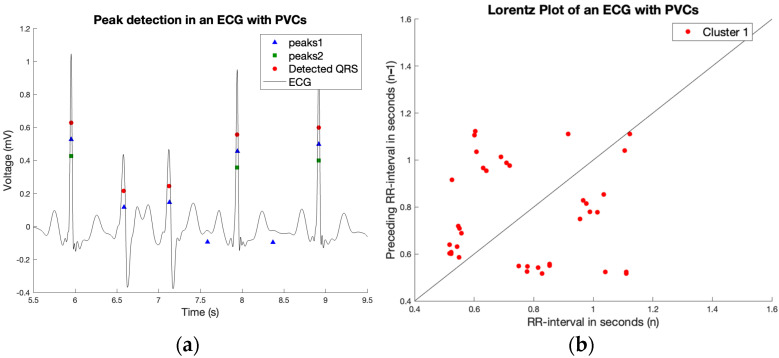
The ECG and found peaks by the algorithm (**a**) of a patient with PVCs, where the clusters were not identified through the algorithm (**b**).

**Table 1 sensors-23-09283-t001:** Baseline patient characteristics.

Variable	Disease/Diagnosis AW	*n* (%)
Cardiac Disease	No disease	173 (24%)
	Atrial Fibrillation	154 (21%)
	Atrial Flutter/Atrial Tachycardia	33
	Ventricular Tachycardia	3
	Junctional Tachycardia	5
	Ventricular Extrasystole	54
	Atrial Extrasystole	21
	First-degree AV-block	77
	Second/third-degree AV-block	21
	Sick Sinus Syndrome/Sinus Bradycardia	65
	Pacemaker	26
	CRT	13
	Right Bundle Branch Block	54
	Left Bundle Branch Block	47
	Intermittent Bundle Branch Block	13
	Left Anterior Hemiblock	23
	Right Heart Axis	13
	Wolff-Parkinson-White Syndrome	26
	Brugada Syndrome	13
	Arrhythmogenic Right Ventricular Cardiomyopathy	20
	Hypertrophic Cardiomyopathy	10
	Long QT Syndrome	8
	Q wave	20
	ST elevation/depression	54
	Negative T	59
AW diagnosis	Sinus Rhythm	455 (62%)
	Atrial Fibrillation	137 (19%)
	Inconclusive	142 (19%)

**Table 2 sensors-23-09283-t002:** The medians for the two tried irregularity features for different ECG abnormalities. The first one (a), called count RR, where the number of regular *RR* intervals with a range of 15 ms are considered and then divided by the heart rate. The second one (b), called the SVD ratio, is the ratio between the two singular values of the Lorenz plot of *RR* intervals.

Disease (*n*)	Count *RR* (a) Median	SVD Ratio (b) Median
Atrial Fibrillation (29)	0.059	7.45
AV block type 1 (12)	0.229	44.30
AV block type 2–3 (4)	0.267	48.42
No abnormalities (35)	0.269	64.41
Premature Atrial Contractions (5)	0.188	7.36
Premature Ventricular Contractions (8)	0.154	15.80
Sick Sinus Syndrome (12)	0.253	53.51

**Table 3 sensors-23-09283-t003:** Comparison table of validation studies reporting sensitivity and specificity of the Apple Watch ECG to identify AF.

	Apple [12]	Abu-Alrub et al. [9]	Racine et al. [10]	Current Study
**Subjects (*n*)**	602	200	734	144
**Sensitivity (%)**	85	87	83	90
**Specificity (%)**	91	86	79	92
**No diagnosis (%)**	9	13	19	0

## Data Availability

Data will be made available after reasonable request.

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
