# Peer review of "Improving Automatic Smartwatch Electrocardiogram Diagnosis of Atrial Fibrillation by Identifying Regularity within Irregularity"

_sensors, 2023, doi:10.3390/s23229283_

Round 1

Reviewer 1 Report

Comments and Suggestions for Authors

The manuscript "Improving automatic smartwatch electrocardiogram diagnosis 2 of atrial fibrillation by identifying regularity within irregularity" presents a refined algorithm for the detection of atrial fibrillation using smartwatch recorded ECG signals. The proposed approach selects candidate AF traces by considering those showing irregularity based on the median RR interval divided by the heart rate and SVD ratio on Lorenz plots of the RR intervals. Values below two thresholds identified through a ROC curve are considered indicative of irregularity and hence candidate AF traces.

In a second step a k-means clustering algorithm is applied to the Lorenz plots of the selected irregular traces and the finding of other clusters indicating regularity of RR intervals was used to classify those traces as non-AF.

Major comments

The introduction needs to provide a broader overview of the literature on automatic identification of AF and on the use of smartwatch ECG for heart diagnosis, with more citations also regarding other groups' work.

The materials and methods section needs to be more precise in defining the used algorithms (SVD, Lorenz plots etc) and appropriate citations need to accompany them. The current explanations of the two figures considered: "The first method included every regular RR interval, which is the median of the RR intervals of the ECG with a range of 15 milliseconds, this is then divided by the heart rate."  as well as "The singular values that resulted from this function are divided and so this gave a ratio in singular values [15,16]. " are incomprehensible and must be rewritten. Please specify the proposed computations using formulas.

Minor comments

The simultaneous recording of the 12-lead electrocardiogram and of the smartwatch ECG would have a better methodological choice, allowing the algorithm and the expert electrophysiologist to evaluate the same data. Why did the authors follow a different approach? 

Please use a larger font for all the text and the axis labels in figures 1 through 4.

l. 156 "where RR intervals within a certain range were divided..." please specify which certain range was considered.

l. 168 The optimal threshold for the RR count is specified as 0.146 while on line 198 the value of 0.158 is used, please clarify.

l. 198 Is the sentence "the cluster count needed to be equal to 0" correct? Isn't there always at least one cluster? 

Comments on the Quality of English Language

The paper is somewhat difficult to read and I suggest some editing by a native english speaker. Please find below a few specific suggestions.

Both in the abstract (l. 14 and 18) and in the text (l. 43) the authors use the term "registration" or "register", where I believe "recording" and "record" are intended.

l. 274 "adjudicate the diagnosis" should probably be "assign" or "define"?

l. 285-288. The sentences "which is not a reference to reality" and "a data set that is more of a reflection of reality" should be improved.

l. 293 "patients... are continuously being detected for arrhythmias" should probably be "monitored" or "inspected"?

Reviewer 2 Report

Comments and Suggestions for Authors

The manuscript presents a novel approach for the automatic diagnosis of atrial fibrillation (AF) by exploring the regularity within irregular ECG patterns. Overall, the paper addresses an important and relevant problem in the field of cardiology and medical signal processing, and it presents a unique approach to automatic AF diagnosis. The combination of RR interval analysis and Lorenz plot-based features is innovative and demonstrates the authors' commitment to exploring novel methods for improved diagnosis.

However, I have the following observations:

1.      The paper lacks a thorough validation of the quality of ECG records. Authors can conduct signal quality assessment for original ECG records using signal quality indices like basSQI and pSQI.

2.      The discussion about the trade-off between inclusivity and specificity is interesting and relevant to the field of AF detection. The authors claim that their algorithm is more inclusive while maintaining high specificity. It would be beneficial to provide additional context or examples to support this claim and address the potential clinical implications of this inclusivity.

3.      In the paper, provide a justification for why k-means clustering was chosen over alternative clustering methods, such as hierarchical clustering, DBSCAN, or Gaussian mixture models.

4.      Consider conducting performance analysis where you apply multiple clustering algorithms to the data and compare their performance in identifying irregular patterns indicative of AF. This could provide valuable insights into the robustness of your approach and help justify the choice of k-means clustering.

5.      Examine existing studies or literature applying machine learning (ML) methods to ECG data for atrial fibrillation (AF) detection. Summarize their findings, highlight their limitations, and provide a comparative analysis with your results. This will provide valuable context for understanding how your algorithm compares to ML-based approaches and enable an assessment of whether your approach outperforms ML-based methods.

6.      The conclusion section should be developed more comprehensively. Consider expanding the content with additional sentences to provide a more thorough summary of the study's key findings and their implications. This will enhance the overall completeness of the conclusion.

Comments on the Quality of English Language

Minor editing required

Reviewer 3 Report

Comments and Suggestions for Authors

This paper proposed a novel algorithm for smartwatch to detect Atrial Fibrillation. Like other similar algorithms, the first step was the detection of the RR interval, then using Lorenz plots to exclude other regular arrhythmias but not Atrial Fibrillation results. According to the experimental results of the paper, compared to the Apple Watch, the algorithm could reduce the number of uncertain results and false positives. Although the work seems to be effective, major issues as below should be considered.

1) How to define the threshold value of ROC? For other arrhythmias or patients, what is different about ROC? It’s good to show the ROC of different arrhythmias.

2) Whether the process of ‘Data arrangement’ is reliable or not? It should be introduced in more detail.

3)In ‘Finding regularity in irregularity’ section, there are just two types of arrhythmias Lorenz plots, but there are more types in the Table 1, how do the other types display on the Lorenz plots?

Comments on the Quality of English Language

The minor issues with the paper include:

1) There are two “irregular” in Abstract.

2) In Introduction, the expression is not rigorous, Atrial Fibrillation is just one of the most common types of cardiac arrhythmia, not the most common.

Round 2

Reviewer 1 Report

Comments and Suggestions for Authors

The authors have answered all my concerns and the paper, with also the replies to the other reviewers, was greatly improved.

Please redraw figure 5 using only 4 of 5 seconds of data, another color other than green for peaks 1 e.g. blue, and more different colors for peaks 2 and Detected QRS. Also please use a different symbol for each series (e.g. o,+,x).

Comments on the Quality of English Language

Some minor English errors are still present.

Reviewer 2 Report

Comments and Suggestions for Authors

The manuscript has been revised carefully. Still, I have the following observations:

1.      The justification for selecting the k-means clustering over alternative clustering methods is not satisfactory.

2. The discussion on the existing studies on the automatic detection of atrial fibrillation (AF) has not been covered thoroughly, including their findings and limitations. However, a performance comparison table might help justify the improvements of automatic AF detection by the proposed approach.

Comments on the Quality of English Language

Moderate editing of English language required

Reviewer 3 Report

Comments and Suggestions for Authors

no more questions or comments on thie ms, thanks.

Comments on the Quality of English Language

no more comments on the language .
